# Surgical Staging in Locally Advanced Cervical Cancer: Precision, Risks, and the ‘Helmet’ Analogy

**DOI:** 10.3390/cancers17213487

**Published:** 2025-10-30

**Authors:** Mikel Gorostidi, Martina Ángeles, Blanca Gil-Ibáñez, Arantxa Lekuona, Alejandra Martinez, Ignacio Zapardiel

**Affiliations:** 1Obstetrics and Gynecology, Hospital Universitario Donostia (San Sebastián, Gipuzkoa), 20014 Donostia-San Sebastián, Spain; 2Biogipuzkoa Health Research Institute, 20014 Donostia-San Sebastián, Spain; 3Department of Medical and Surgical Specialties, Faculty of Medicine and Nursing, University of the Basque Country (UPV/EHU), 48940 Leioa, Spain; 4Gynecologic Oncology Unit, Vall d’Hebron Barcelona Hospital Campus, Universitat Autònoma de Barcelona, 08035 Barcelona, Spain; martina.angeles@vallhebron.cat; 5Gynecologic Oncology and Minimally Invasive Surgery Unit, Gynecology and Obstetrics Department, University Hospital 12 de Octubre, 28041 Madrid, Spain; 6Research Institute i+12, 28041 Madrid, Spain; 7Université de Toulouse Oncopole Claudius Regaud, Institut Universitaire du Cancer Toulouse Oncopole, 31100 Toulouse, France; 8CRCT UMR 1037 INSERM, 31100 Toulouse, France; 9Gynecologic Oncology Unit, La Paz University Hospital, 28046 Madrid, Spain; ignacio.zapardiel@uam.es

**Keywords:** uterine cervical neoplasms, lymph node excision, para-aortic lymph nodes, neoplasm staging, positron-emission tomography and computed tomography, chemoradiotherapy, biomarkers, tumor, minimally invasive surgical procedures

## Abstract

**Simple Summary:**

Locally advanced cervical cancer often spreads to lymph nodes near the aorta (para-aortic nodes). Today, PET/CT scans are the standard way to look for this spread, but they frequently miss very small deposits. Surgeons can remove and examine these nodes (surgical staging), which is the most accurate method to uncover hidden disease. However, operating on everyone is not always necessary or practical, and surgery must not delay the start of chemoradiation. This review explains when surgical staging may help—mainly in carefully selected, higher-risk patients—by guiding whether to extend radiation fields or adjust systemic therapy. We summarize the best available studies, including one randomized trial, large databases, and ongoing prospective trials. We also discuss new tools that could sharpen selection and reduce morbidity, such as sentinel node mapping, molecular assays (e.g., OSNA), and blood-based HPV-DNA biomarkers. Overall, we propose a pragmatic, quality-assured pathway led by accredited centers: combine advanced imaging, selective surgery, and biomarkers so that the patients most likely to benefit are the ones who receive surgical staging.

**Abstract:**

**Background/Objectives**: This study aims to critically appraise the role of para-aortic surgical staging in locally advanced cervical cancer (LACC) in the era of advanced imaging, and to outline how selective surgery and biomarkers could be integrated within modern, quality-assured treatment pathways. **Methods**: Narrative review of randomized trials, large databases, and prospective/retrospective series comparing para-aortic lymphadenectomy with imaging-based staging; focused appraisal of Uterus-11, NCDB analyses, and ongoing prospective trials (PAROLA with Senti-PAROLA as one of its sub-studies and PALDISC). Emerging technologies (PET/MRI, radiomics/AI) and molecular assays (OSNA, HPV-ctDNA) were also assessed. **Results**: PET/CT remains the standard for distant staging, but sensitivity for low-volume nodal disease (<5 mm) is poor; in pelvic-positive/para-aortic-negative patients, occult para-aortic metastases approach ~21%. Para-aortic surgical staging modifies radiotherapy planning in ~18% of cases and can act as a de-escalation tool by avoiding unnecessary extended-field CRT (EF-CRT) when para-aortic nodes are negative. Uterus-11 showed no overall survival difference versus CT-based staging, but suggested benefit in FIGO 2009 stage IIB; its design (CT comparator, optimistic assumptions, limited power) constrains inference. Minimally invasive extraperitoneal/transperitoneal staging is feasible with low morbidity in expert centers, yet real-world management may worsen outcomes. The role of systemic intensification in node-positive disease remains undefined: PALN-positive patients were excluded from the INTERLACE trial. In the KEYNOTE-826 study, subgroup analyses according to nodal status were not reported, although the benefit of pembrolizumab remained consistent irrespective of bevacizumab use. Sentinel para-aortic mapping and biomarkers (e.g., HPV-ctDNA) may refine selection and reduce morbidity. **Conclusions**: Surgical staging is the most accurate method to detect occult para-aortic disease. Its routine use is not justified, but it may benefit selected high-risk patients, particularly where decisions on EF-CRT or systemic therapy hinge on para-aortic status. Future practice should integrate advanced imaging, selective surgery, and biomarkers within accredited centers, guided by large collaborative trials conducted under international quality frameworks such as ESGO/ESTRO/ESP guidelines.

## 1. Introduction

Locally advanced cervical cancer (LACC) continues to represent a major therapeutic challenge, despite advances in concurrent chemoradiation and imaging technology. PET/CT is currently the most widely used tool for distant staging, with high accuracy in detecting nodal and systemic dissemination. However, the sensitivity of PET/CT decreases dramatically for small-volume nodal metastases (<5 mm), reaching only about 14% [1,2,3]. For deposits measuring 5–10 mm, sensitivity is intermediate, in the range of 40–50% [4,5]. In the same series, sensitivity increased progressively with nodal size, achieving high values of approximately 90% for metastases > 10 mm [6,7].

Reported sensitivity for overall nodal disease ranges from ~20% to 80% depending on volume and site [6,8,9], with the risk of false negatives being particularly high in cases with positive pelvic nodes but negative para-aortic uptake, and especially in the para-aortic area.

In this context, surgical staging remains the most accurate method to identify occult nodal disease. The fundamental question is whether it should still be recommended systematically, or whether its role should be reserved for carefully selected groups of patients. Importantly, we still do not know whether extended-field chemoradiotherapy (EF-CRT) improves survival compared with whole-pelvis radiotherapy (including the common iliac regions, as is usually delivered in pelvic N+ cases). To illustrate this dilemma, the analogy of the helmet for motorcyclists may be useful: differences in survival are evident only among those who suffer an accident; likewise, surgical staging primarily provides prognostic information in those patients with occult nodal involvement undetected by imaging. It is well established that patients with para-aortic metastases have a markedly worse prognosis; what remains uncertain is whether accurately identifying them and adapting treatment accordingly can ultimately improve their outcomes.

Undoubtedly, in an ideal scenario, imaging techniques could achieve such precision in approximating surgical diagnosis that, in this context, surgery would lose its added value. However, to date, we have not yet reached that point. Emerging technologies such as PET/MRI may improve sensitivity and are promising for detecting small-volume nodal disease. However, there is still no evidence of clinical superiority over PET/CT; these techniques remain investigational and do not resolve the sensitivity problem [10]. Table 1. We know that PET/CT still underestimates nodal disease. A recent meta-analysis reported an upstaging rate of 12% (95% CI 8–16) with surgical staging, and in the subgroup of pelvic-positive but para-aortic-negative uptake patients the risk of occult para-aortic disease reached 21% (95% CI 17–26%) [11]. This group represents precisely the population most likely to benefit from surgical staging, and is the focus of the ongoing PAROLA trial [12], although not necessarily the only one in whom surgical staging might be relevant. The next leap will be to combine advanced imaging with radiomics and artificial intelligence to identify the patients with higher risk of occult nodal involvement [13].

Although this review was conceived as a narrative rather than a systematic analysis, the literature search was structured and comprehensive. Relevant studies were identified through targeted PubMed/MEDLINE searches using predefined terms (‘cervical cancer,’ ‘para-aortic staging,’ ‘lymphadenectomy,’ ‘PET/CT,’ ‘chemoradiotherapy,’ and ‘surgical staging’), with additional manual screening of reference lists to ensure inclusion of all major randomized trials, meta-analyses, and large multicenter series published up to June 2025. This approach aimed to provide a balanced overview of current evidence and to minimize selection bias.

At this juncture, the present paper seeks to highlight both the strengths and the limitations of surgical staging and imaging-based staging, in order to foster a more critical appraisal of where we currently stand.

## 2. Available Evidence and Its Limitations

Most of the available evidence derives from retrospective series (level IV). To date, only two randomized trials have addressed this question: the Uterus-11 study [14], which compared CT-based with surgical para-aortic staging, and the earlier trial by Lai et al. [15], whose interpretation is hampered by important limitations (small sample size, treatment delays, surgical morbidity, technological obsolescence, and heterogeneous surgical approaches). Importantly, Uterus-11 suffers from major design flaws. The trial assumptions implied very low survival in the control arm and a large treatment effect for surgical staging, which were inconsistent with contemporary benchmarks—for instance, 5-year overall survival around 70% in modern chemoradiotherapy series with MRI-guided brachytherapy (EMBRACE-I) [16] and a modest absolute survival gain (~6%) historically attributable to concurrent chemoradiation over radiotherapy alone in Cochrane analyses [17]. As a result, Uterus-11 was likely underpowered and relied on CT rather than PET/CT for clinical staging, without specifically enriching for a high-risk population such as FIGO 2018 IIIC1.

Unsurprisingly, in the overall cohort of 255 patients, no statistically significant differences were observed in disease-free survival (DFS) (HR 0.71, 95% CI 0.48–1.05, *p* = 0.084) or overall survival (OS) (HR 0.69, 95% CI 0.48–1.05, *p* = 0.096), although a favorable trend was noted in the surgical arm, suggesting the possibility of a type II error. Subgroup analyses are more intriguing: in stage II patients, surgical staging was associated with significant improvements in DFS (HR 0.51, 95% CI 0.30–0.86, *p* = 0.011) and cancer-specific survival (CSS) (HR 0.61, 95% CI 0.40–0.93, *p* = 0.020). These exploratory results, however, must be interpreted with caution, as they derive from post hoc analyses. Although formally negative, Uterus-11 nonetheless underscores the need for a modern, adequately powered, collaborative trial in well-defined high-risk populations, using current imaging standards.

Key lessons from the Uterus-11 trial:First and only randomized trial of surgical vs. clinical staging in LACC.No significant survival benefit in the overall cohort (DFS HR 0.71, *p* = 0.084; OS HR 0.69, *p* = 0.07).Trend toward improved outcomes in the surgical arm, suggesting possible underpowering.Subgroup analysis (stage IIB) showed significant DFS and CSS benefits (post hoc; interpret with caution).Main limitations: use of CT instead of PET/CT as comparator, insufficient statistical power.Implication: Although formally negative, results support the rationale for larger randomized/international trials (e.g., PAROLA).

A more recent large retrospective study from the US National Cancer Database evaluated 3540 patients (2010–2015), of whom only 333 (9.4%) underwent surgical staging [18]. After adjustment, no significant differences in 4-year OS were observed (HR 1.07) between surgical and imaging staging. Notably, surgical staging detected para-aortic involvement in a higher proportion of patients compared with imaging alone (27.3% vs. 13.2%, *p* < 0.001), potentially allowing more appropriate use of EF-CRT. Yet, the study suffers from several important limitations: details regarding radiation fields, doses, nodal boosts, brachytherapy protocols, and chemotherapy regimens were lacking, all of which are major determinants of outcome. Furthermore, the decline in surgical staging rates during the study period may in part reflect the increasing use of prophylactic para-aortic irradiation, a strategy that other series have linked to reduced para-aortic failures and improved survival in patients with pelvic and common iliac nodal disease. Importantly, outcomes were not reported separately for stage FIGO 2018 IIIC2, the subgroup most likely to benefit [19]. Interestingly, survival curves diverged beyond 6 years, raising the possibility that longer follow-up or a larger cohort might have demonstrated a benefit. Ultimately, the true impact of surgical staging on oncological outcomes can only be clarified by a modern phase III randomized trial with adequate statistical power, such as the ongoing PAROLA study, which specifically evaluates 3-year DFS in stage IIIC1 patients staged with pretreatment PET/CT (Table 2).

Lymphatic dissemination in cervical cancer generally follows a hierarchical and predictable pattern, progressing from pelvic to common iliac and para-aortic nodes. Isolated para-aortic involvement in the absence of pelvic nodal disease is exceedingly uncommon, typically below 3% of cases. Several large series have confirmed that the probability of para-aortic metastasis increases proportionally with both the size and the number of involved pelvic nodes, supporting the concept of stepwise spread. In the multicenter prospective study by Gouy et al. [24], patients presenting with bulky pelvic nodes (>10 mm) showed para-aortic involvement rates approaching 25%. Similar findings were reported by Martinez et al. [9] and Jiang et al. [22] in contemporary PET/CT-guided cohorts. This evidence reinforces a selective, risk-adapted approach to surgical staging—focusing on patients with a high pelvic nodal burden or radiologic suspicion of common iliac extension—rather than a systematic application in all cases.

From both Uterus-11 and retrospective cohorts, several points emerge:In stage IIB patients, surgical staging may provide significant survival benefit.The global effect across all IB2–IVA patients is diluted, masking subgroup advantages.Para-aortic lymphadenectomy modifies treatment plans in ~18% of cases, and up to 44% in some series [20,21,22].

It is also worth noting that, under the former FIGO 2009 classification, stage II accounted for approximately 70% of all “locally advanced” cases, as the system was purely clinical. This raises the possibility that patients with less locally extended disease (stage II) but nodal involvement might hypothetically benefit the most from surgical staging, particularly those with low-volume para-aortic metastases. In such patients, accurate identification of occult disease could allow more tailored treatment—avoiding both undertreatment and unnecessary extended-field irradiation—whereas in more advanced local stages the relative impact of nodal staging may be diluted by bulky pelvic disease.

A comprehensive review by Martínez and colleagues [23] further emphasized these issues, underscoring that PET/CT has limited sensitivity for infra-radiological para-aortic disease, that systematic EF-CRT may expose up to three quarters of patients to unnecessary extended-field irradiation, and that adequately powered randomized trials—such as PAROLA—are needed to clarify the true therapeutic value of surgical staging.

## 3. Safety, Timing and Real-World Considerations

The main concern is whether surgical staging delays the initiation of chemoradiotherapy, potentially affecting prognosis. However, several recent prospective series have demonstrated that, when performed via a minimal invasive approach, either trans or extraperitoneal, in expert centers, para-aortic lymphadenectomy shows minimal morbidity and does not significantly delay treatment initiation. For instance, Puga et al. (2022) found laparoscopic surgical staging both safe and reproducible, enabling a reduction in radiotherapy volumes in almost half of patients with LACC [21]. He et al. (2023) similarly report that extraperitoneal pelvic and para-aortic lymphadenectomy allows accurate abdominal nodal assessment and better treatment planning [25]. Furthermore, Nasioudis et al. (2023) note that in high-volume centers, such procedures do not result in clinically significant delays to chemoradiation, while identifying occult disease in up to 25% of patients [26]. Petitnicolas et al. (2017) also support the feasibility of this approach without delaying definitive therapy [27]. With optimized logistics (dedicated slots, rapid pathology reporting, multidisciplinary coordination), the interval to treatment can be preserved.

However, in real-world practice—including in reference centers—the situation could be different. Even with efficient organization, any additional procedure carries a risk of delay, and prolonged time to treatment initiation is known to negatively impact survival in cervical cancer. In fact, a recent meta-analysis demonstrated that a 4-week delay increases mortality by 27% (HR 1.27; 95% CI 1.12–1.45) [28]. In addition, observational data indicate that delays of 4 months or more roughly double the risk of death (HR 2.31) [29]. Some series also suggest that waiting more than 60 days may be associated with poorer outcomes [30]. Under optimal conditions, treatment can start within 10–21 days after surgery, as reported in the UTERUS-11 trial [31].

Furthermore, although extraperitoneal lymphadenectomy is generally safe, it is not morbidity-free: intraoperative complications, large lymphocysts, recurrent infections, lymphedema, ureteral obstruction, and urinomas have all been reported [32]. Among late adverse events, lower-limb lymphedema—although uncommon after isolated extraperitoneal para-aortic staging—represents the most troublesome and persistent morbidity when pelvic dissection is added, with substantially higher long-term rates reported in combined pelvic ± para-aortic lymphadenectomy series; in contrast, after isolated para-aortic staging the main morbidity consists of lymphoceles and chylous ascites.

Thus, two conditions must be met: surgical staging must not add significant morbidity, and it must not delay chemoradiation. Only when these requirements are satisfied can surgical staging be reasonably considered, as it may potentially provide a benefit for selected patients.

Beyond clinical outcomes, the applicability of surgical staging must also be interpreted in light of real-world constraints. When considering *real-world* practice, it is important to acknowledge that the feasibility of surgical staging is highly dependent on local resources and health system capacity. In several high-volume centers in regions such as South America, the number of patients diagnosed with LACC each year largely exceeds the logistical possibilities of performing systematic para-aortic staging in all cases. In these settings, a more pragmatic approach based on careful patient selection may be appropriate, allowing potential benefits of surgical staging to be captured in subgroups of patients while avoiding the impracticality of universal application.

In parallel, the availability of diagnostic imaging also influences the feasibility of surgical staging. While PET/CT remains the recommended standard for nodal and distant assessment, its accessibility is still limited in many regions worldwide, particularly across low- and middle-income countries. In these settings, contrast-enhanced CT continues to serve as a pragmatic and widely accessible tool for pre-treatment evaluation and surgical decision-making. Although CT lacks the sensitivity of PET/CT for small-volume disease, it can reliably identify enlarged or morphologically suspicious lymph nodes and thus remains clinically valuable for selecting candidates for surgical staging. When intraoperative frozen-section assessment of common iliac nodes is incorporated, this approach can effectively guide the decision on whether to extend dissection to the para-aortic region. Such resource-adapted algorithms may optimize patient selection while maintaining treatment timelines, thereby preserving the balance between diagnostic precision and feasibility in real-world oncology practice.

A relevant example of such a resource-adapted strategy is the institutional practice described by several high-volume centers, where decisions regarding para-aortic lymphadenectomy are guided by CT findings and intraoperative pathological assessment. In this pragmatic algorithm, pelvic or common iliac nodes exceeding 2 cm in size prompt consideration for lymphadenectomy, whereas in patients with smaller para-aortic nodes (<1 cm), intraoperative frozen-section examination of the common iliac area determines whether to proceed to para-aortic dissection. This stepwise approach exemplifies an effective compromise between diagnostic precision, procedural safety, and treatment timeliness in settings where PET/CT is not routinely accessible.

Surgical staging should be restricted to accredited reference centers, with expertise in low-prevalence, high-complexity gynecologic cancers, systematic outcome monitoring, and capacity to minimize morbidity and avoid treatment delays. Patient selection is critical, particularly considering comorbidities and frailty. These reference centers must be properly accredited, capable of fully implementing ESGO/ESTRO/ESP guidelines, consistently meeting established surgical and radiotherapy quality indicators, and possessing particular expertise in endoscopic gynecologic oncological procedures, which is crucial to ensure safety and minimize morbidity [33,34].

The ongoing PALDISC trial (NTR4922) has been designed to prospectively evaluate the feasibility, safety, and diagnostic yield of para-aortic lymphadenectomy in locally advanced cervical cancer [35,36]. The study includes women with FIGO 2009 stage IIB–IVA cervical cancer scheduled for primary chemoradiation and aims to compare histologic findings with PET/CT in terms of sensitivity for occult para-aortic disease. Recruitment started in 2014, but to date, no peer-reviewed results or formal congress communications have been published. Its primary endpoints are perioperative morbidity, potential delays in treatment initiation, and concordance between surgical and imaging staging. Although no data are available yet, PALDISC is expected to provide essential information on the real-world applicability of surgical staging and to guide the design of adequately powered phase III studies focused on oncologic outcomes. Nonetheless, the low planned sample size of only 30 patients will inevitably provide limited information in this regard. This study should not be confused with the PAROLA trial [12], which aims to recruit 510 patients; unlike PALDISC, PAROLA specifically focuses on women with a higher likelihood of para-aortic involvement, FIGO 2018 stage IIIC1, thereby enriching the population most likely to benefit from tailored treatment strategies. The trial aims to evaluate whether chemoradiation with a tailored external beam radiation field based on surgical para-aortic staging is associated with improved 3-year DFS and is currently actively recruiting.

## 4. The “Helmet” Analogy

The debate around the global usefulness of surgical staging can be misleading. If we compare overall mortality among all motorcyclists with and without helmets, we may find no significant differences. The benefit becomes evident only in those who suffer an accident, and even then, not all will survive thanks to the helmet. In this analogy, the patient “wearing a helmet” is the one who undergoes lymphadenectomy, whereas the patient staged only by imaging corresponds to the rider without a helmet. The actual risk of falling—analogous to the likelihood of nodal involvement—depends on factors such as speed, road conditions, and weather, which in our setting translate into FIGO stage and other clinical or radiological risk factors (Figure 1).

It is well established that the prognosis of patients with para-aortic nodal involvement is particularly poor, especially when metastases exceed 5 mm. What remains uncertain, however, is whether optimal management of these patients can translate into improved outcomes. This represents the fundamental question that must be addressed before advocating for a more precise diagnostic strategy at the expense of potential treatment delays and increased morbidity. In this context, it is worth mentioning the Spanish multicenter study from the SEGO Spain-GOG group [37], which evaluated laparoscopic extraperitoneal para-aortic staging in 513 patients with locally advanced cervical cancer. The study confirmed the feasibility and safety of the technique and demonstrated its utility in tailoring radiotherapy fields according to histological nodal status. However, no significant differences in survival outcomes were observed compared with the control group, in which EF-CRT was routinely applied without surgical staging. Importantly, para-aortic lymphadenectomy may also help avoid overtreatment with EF-CRT in patients without para-aortic metastases [23]. Nevertheless, with the advent of modern radiotherapy techniques such as Intensity-Modulated Radiotherapy (IMRT)/Volumetric Modulated Arc Therapy (VMAT), the toxicity profile of EF-CRT has substantially improved. In contemporary series, grade ≥ 3 toxicities are reported around ~9% for gastrointestinal and ~3% for genitourinary events, with grade ≥ 3 hematologic toxicity ~10–16%, particularly when concurrent chemotherapy is administered [38,39]. These findings suggest that the therapeutic value of para-aortic staging may rely on the background radiotherapy strategy, and that its prognostic impact remains to be fully clarified.

Another important consideration is the role of EF-CRT and systemic treatment in patients with nodal involvement. Current international guidelines list EF-CRT as optional, and practice varies widely: in some centers, EF-CRT is systematically offered to all FIGO 2018 stage IIIC1 patients, even though para-aortic nodal involvement is not confirmed [39]. In this context, para-aortic lymphadenectomy could serve as a “de-escalation tool,” potentially avoiding up to 75% of unnecessary EF-CRT. Importantly, the safety profile of EF-CRT has improved with modern techniques such as IMRT and VMAT, where recent series report grade ≥3 gastrointestinal toxicity around 9%, genitourinary ~3%, and hematologic ~10–16% when combined with concurrent chemotherapy [40].

The integration of systemic therapies in node-positive disease remains uncertain. In the INTERLACE trial, patients with para-aortic involvement were excluded [41,42]. In KEYNOTE-826, approximately 20% of the cohort presented para-aortic disease, but no subgroup analyses according to nodal status have been reported [26,43,44]. The only subgroup analyses available focused on the use of bevacizumab, demonstrating that the benefit of pembrolizumab was consistent regardless of anti-angiogenic therapy. This highlights a critical evidence gap: para-aortic positive patients, who are at the highest risk of systemic spread, remain the very group most likely to benefit from treatment intensification, yet their outcomes with immune checkpoint blockade are still undefined. Surgical staging may therefore not only guide radiotherapy fields but also inform the rational adaptation of systemic strategies. This is explicitly recognized in the PAROLA trial [12], where para-aortic status constitutes a stratification factor, and prespecified subgroup analyses will provide much-needed data on the potential benefit of tailored systemic therapy in this population.

## 5. Refining Surgical Staging: From OSNA to Biomarkers

New molecular intraoperative techniques such as the one-step nucleic acid amplification (OSNA) assay, which quantifies CK19 mRNA in lymphatic tissue, have shown high concordance with ultrastaging in sentinel lymph node assessment for cervical cancer [45,46,47]. OSNA is particularly sensitive for detecting low-volume disease, including micrometastases, and thus could theoretically be applied to para-aortic lymphadenectomy specimens to improve identification of occult metastases. While currently investigational and not yet incorporated into clinical guidelines, this approach may enhance staging accuracy by reducing false negatives in patients with low-volume nodal involvement.

Some authors, such as Gouy and colleagues, have reported that patients with para-aortic nodal involvement measuring ≤5 mm—infra-radiological metastases, which they termed “micrometastases”—achieved survival outcomes comparable to those without para-aortic spread, suggesting a potential therapeutic benefit of surgical staging in this subgroup and a possible role for nodal debulking [24,48]. However, these findings were based on only 13 patients, which makes it difficult to draw firm conclusions. This definition differs from the current pathological classification of micrometastasis (pN1mi), which refers to deposits >0.2 mm and ≤2 mm detected by ultrastaging. This discrepancy underscores the importance of consistent use of terminology when evaluating the prognostic impact of low-volume nodal disease in cervical cancer.

Moreover, an important added value of sentinel node detection at the para-aortic level lies in the potential to minimize, or even avoid, the morbidity associated with a full surgical staging procedure. This concept is currently being evaluated in the Senti-PAROLA sub-study [12], which aims to determine the sensitivity, specificity, and predictive values of para-aortic sentinel nodes compared with complete lymphadenectomy for the detection of occult metastases. The incorporation of a more precise surgical approach that avoids extensive lymphadenectomy and its associated morbidity, without compromising diagnostic accuracy, would represent another step forward in the era of high-precision gynecologic oncology.

The critical threshold—and the Achilles’ heel of PET/CT—remains nodal metastases < 5 mm, where detection is particularly poor. This limitation underpins the continued interest in surgical staging and in complementary techniques such as ultrastaging or molecular assays like OSNA, which are capable of identifying low-volume diseases more reliably.

The integration of molecular biomarkers may represent the next frontier in the personalized management of locally advanced cervical cancer. Circulating HPV-DNA (HPV-ctDNA) has shown promising correlation with tumor burden, treatment response, and minimal residual disease, potentially serving as a dynamic marker of occult dissemination beyond the pelvic field. Early studies suggest that pre-treatment HPV-ctDNA positivity and post-treatment persistence are associated with an increased risk of para-aortic and distant relapse, supporting its potential role in identifying patients who might benefit from extended-field chemoradiotherapy or systemic intensification [49,50].

Similarly, the OSNA assay—by providing real-time, quantitative assessment of cytokeratin 19 mRNA in sentinel lymph nodes—offers an objective molecular surrogate for nodal tumor load, reducing interobserver variability and allowing intraoperative decision-making. Beyond confirming micrometastases, molecular quantification could in the future support risk models predicting non-sampled para-aortic involvement. Combining OSNA with radiomics or ctDNA profiles could enable hybrid algorithms integrating surgical, imaging, and molecular data to guide radiotherapy field tailoring with higher precision.

Beyond HPV-ctDNA, additional classes of circulating biomarkers are gaining attention. Specific microRNAs (e.g., miR-21, miR-92a, and miR-205) have been shown to correlate with tumor burden, response to chemoradiation, and recurrence risk, suggesting their utility as dynamic indicators of minimal residual disease. Likewise, methylation markers—particularly promoter hypermethylation of tumor suppressor genes such as CADM1, MAL, or FAM19A4—are emerging as robust epigenetic signatures of high-grade disease and may refine post-treatment surveillance when combined with ctDNA assays. Integrating these molecular layers could enable a composite liquid biopsy profile capable of capturing both tumor genetic activity and epigenetic reprogramming, advancing precision follow-up strategies in locally advanced cervical cancer [51,52].

These advances will require validation within prospective, biomarker-driven trials conducted under standardized quality frameworks, as endorsed by ESGO/ESTRO/ESP

## 6. Conclusions

Surgical staging remains the most accurate method to detect occult para-aortic disease. However, its routine use is not justified in all patients, given the associated morbidity and the improved accuracy of modern imaging techniques. The key lies in identifying those at highest risk, for whom treatment decisions regarding extended-field chemoradiotherapy or systemic therapy depend critically on para-aortic status. Selective use within accredited centers, guided by multidisciplinary consensus and evidence-based algorithms, can maximize the therapeutic benefit while minimizing harm.

Future management should integrate advanced imaging, selective surgical staging, and emerging molecular biomarkers to achieve a more precise and individualized approach. Large collaborative studies conducted under international quality frameworks, such as the ESGO/ESTRO/ESP guidelines, are essential to validate these strategies and ensure their feasibility and equity in diverse healthcare settings.

## Figures and Tables

**Figure 1 cancers-17-03487-f001:**
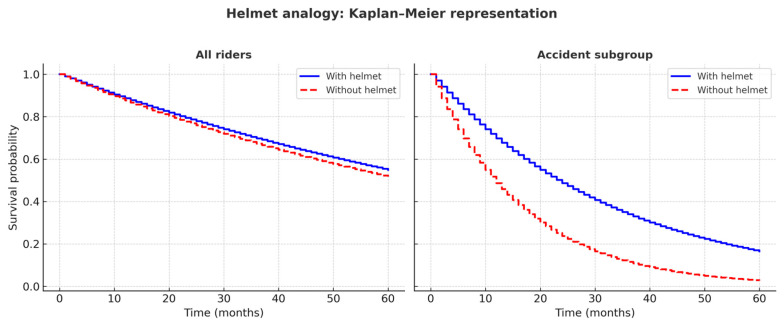
Hypothetical Kaplan–Meier curves illustrating the “helmet analogy.” When considering all riders, survival appears similar regardless of helmet use. However, in the subgroup of riders involved in an accident, survival is markedly better among those wearing a helmet. This figure was created for illustrative purposes only and does not represent real patient data.

**Table 1 cancers-17-03487-t001:** Sensitivity of PET/CT for nodal metastases in cervical cancer according to deposit size.

Nodal Deposit Size	Approximate PET/CT Sensitivity	References
<5 mm	10–15%	Kitajima 2008 [1]; Chou 2006 [3]
5–10 mm	40–50%	Grigsby 2001 [4]
>10 mm	80–90%	Gouy 2021 [6]

**Table 2 cancers-17-03487-t002:** Key studies on surgical versus imaging staging in locally advanced cervical cancer.

Study/Source	Design and Population	Intervention/ Comparator	Main Findings	Limitations
Uterus-11 Trial (Marnitz 2020) [14]	RCT, n = 255 (FIGO IIB–IVA)	CT vs. para-aortic surgical staging before CRT	Stage II subgroup: improved DFS (HR 0.51, *p* = 0.011) and CSS (HR 0.61, *p* = 0.020). Whole cohort: no significant differences (DFS *p* = 0.084).	Comparator was CT (less sensitive than PET/CT); underpowered; positive results only in post hoc subgroup.
NCDB Study (Nasioudis 2022) [18]	Retrospective cohort, n = 3540 (2010–2015)	Imaging vs. surgery (333 pts, 9.4%)	Para-aortic disease identified in 27% with surgery vs. 13% with imaging. No OS difference at 4 years (HR 1.07, ns); survival curves diverged >6 years.	Non-randomized; no separate IIIC2 subgroup; limited follow-up.
Meta-analysis (Thelissen 2022) [11]	Systematic review and meta-analysis (>2000 pts)	Imaging vs. surgery	Overall upstaging: 12%. In pelvic-positive/para-aortic-negative: 21% risk of occult para-aortic disease.	Predominantly retrospective studies; heterogeneity; possible publication bias.
Multicenter retrospective series (Ramirez 2011 [20]; Puga 2022 [21]; Jiang 2024 [22]; Martinez 2020 [23]	Observational, 100–500 pts/series	Extraperitoneal para-aortic staging vs. imaging	RT planning modified in 18–44% of patients; low morbidity in expert centers.	Level IV evidence; variability in logistics and surgical expertise.

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
