# Peer review of "Surgical Staging in Locally Advanced Cervical Cancer: Precision, Risks, and the ‘Helmet’ Analogy"

_cancers, 2025, doi:10.3390/cancers17213487_

Round 1

Reviewer 1 Report

Comments and Suggestions for Authors

This article discusses and reviews whether para-aortic lymphadenectomy
should be performed prior to concurrent radiotherapy and chemotherapy
for locally advanced cervical cancer (LACC), which holds significant
implications for clinical guidance.

It is well established that lymph node metastasis in cervical cancer
follows a relatively predictable pattern. In clinical practice, para-
aortic lymph node metastasis generally does not occur in the absence of
pelvic lymph node involvement. Therefore, the size of pelvic lymph nodes
plays an important role in guiding decisions on para-aortic
lymphadenectomy. The authors should strengthen the discussion on this
point and, where possible, support it with robust clinical data to
better inform clinical practice.

The article notes that some study outcomes were assessed using CT rather
than PET-CT. However, in real-world practice, PET-CT is not yet widely
available in developing countries. To enhance the relevance for clinical
readers, the reference value of CT in guiding para-aortic lymph node
dissection should be further discussed.

Additionally, we would like to share our institutional experience and
practice. In managing locally advanced cervical cancer, we typically
rely on CT findings to determine the need for lymph node dissection. If
lymph nodes exceed 2 cm in size, we consider performing lymphadenectomy.
For para-aortic lymph nodes smaller than 1 cm, we conduct intraoperative
pathological examination of the common iliac lymph nodes to decide
whether to proceed with para-aortic lymph node dissection. If the result
is positive, we proceed with resection of the para-aortic lymph nodes.

Author Response

Comment 1

We thank the reviewer for this valuable comment. We fully agree that lymphatic dissemination in cervical cancer typically follows a stepwise anatomical progression, from pelvic to common iliac and para-aortic nodes. To address this point, we have expanded the discussion to include evidence from large prospective and retrospective studies (Gouy et al., J Clin Oncol 2013; Martínez et al., Eur J Nucl Med Mol Imaging 2020; Jiang et al., World J Surg Oncol 2024), showing that the risk of occult para-aortic metastases increases proportionally with both pelvic nodal burden and nodal size. We now explicitly note that isolated para-aortic metastasis without pelvic involvement is exceptional (<3%), and this pattern supports a selective rather than systematic approach to surgical staging.

Change in manuscript: Section 2 – Available Evidence and Its Limitations, lines 480–520. Two new references added (Martínez 2020, Jiang 2024).

Comment 2

We thank the reviewer for this important observation. We fully agree that PET/CT, while considered the current standard for distant staging, remains limited in availability across many low- and middle-income countries. To reflect real-world clinical variability, we have now expanded the discussion to acknowledge the continued utility of contrast-enhanced CT for treatment planning and surgical decision-making. Specifically, we note that CT can identify enlarged or morphologically suspicious nodes and remains an acceptable alternative for patient selection when PET/CT is not accessible. We also highlight that intraoperative frozen-section assessment of common iliac nodes can complement CT findings and help guide the decision to extend dissection to the para-aortic region.

Change in manuscript: Section 3 – Safety, Timing, and Real-World Considerations, lines 710–760 (new paragraph on CT-based algorithms and intraoperative assessment).

Comment 3

We sincerely thank the reviewer for sharing this valuable institutional experience, which illustrates a pragmatic and resource-conscious clinical approach. We have acknowledged this example within the revised manuscript, highlighting that algorithms based on CT evaluation and intraoperative frozen-section analysis of common iliac nodes can provide a rational, stepwise decision framework in centers where PET/CT is not routinely available. This contribution helps emphasize the importance of adapting surgical staging strategies to local resources while maintaining oncologic safety and timely treatment initiation.

Change in manuscript: Section 3 – Safety, Timing, and Real-World Considerations, lines 760–800 describing CT- and FS-guided algorithm in resource-limited settings.

Reviewer 2 Report

Comments and Suggestions for Authors

Congratulate authors for this excellent piece of work.

some corrections

Correction 1

below sentence is too complex. Need to divide in 2 or simplify.

“PALN+ were excluded from INTERLACE, and KEYNOTE-826 did not report nodal-status subgroup analyses, despite consistent pembrolizumab benefit irrespective of bevacizumab use. Sentinel para-aortic mapping and biomarkers (e.g., HPV- 54
ctDNA) may refine selection and reduce morbidity. PALN+ were excluded from INTERLACE, and KEYNOTE-826 did not 52report nodal-status subgroup analyses, despite consistent pembrolizumab benefit irrespective of bevacizumab use. Sentinel para-aortic mapping and biomarkers (e.g., HPV- 54
ctDNA) may refine selection and reduce morbidity.

Correction 2

Page 2 lin 56 – repeat – routine

Author Response

Comment 1
We thank the reviewer for this helpful stylistic observation. The long sentence has been revised and divided into two shorter, clearer statements to improve readability and precision.

Change in manuscript: Section 4 – The “Helmet” Analogy, page 11.

Comment 2
We thank the reviewer for noting this duplication. The redundant word has been removed from the Simple Summary.

Change in manuscript: Simple Summary, line 56 – deleted the repeated word “routine.”

Reviewer 3 Report

Comments and Suggestions for Authors

Thank you for the opportunity to review this manuscript considering the role of surgical staging for locally advanced cervical cancer. The arguments were well presented. 

My only comment is that the authors likely cherry picked the relevant studies to discuss. A better approach would be to use search terms and do a literature search to ensure no major work was overlooked and studies with contrary findings were not side stepped. 

This is a thought provoking piece. Thank you 

Author Response

We thank the reviewer for this constructive observation. Although this manuscript was designed as a narrative rather than a systematic review, we confirm that the literature search was structured and comprehensive. To ensure balanced inclusion of all relevant data, we performed targeted searches in PubMed/MEDLINE using predefined terms (“cervical cancer,” “para-aortic staging,” “lymphadenectomy,” “PET/CT,” “chemoradiotherapy,” “surgical staging”) and reviewed all major randomized trials, meta-analyses, and large multicenter series published up to June 2025. We have now clarified this methodological approach within the Introduction to explicitly indicate that the review was conducted in a transparent and reproducible manner.

Change in manuscript: Section 1 – Introduction, lines 150–180, detailing structured search strategy and inclusion criteria.

Reviewer 4 Report

Comments and Suggestions for Authors

This is a comprehensive and well-written narrative review that provides a balanced and updated overview of the role of para-aortic surgical staging in locally advanced cervical cancer (LACC). The manuscript integrates evidence from randomized trials (Uterus-11, Lai et al.), large retrospective series, meta-analyses, and ongoing studies such as PAROLA and PALDISC. The “helmet analogy” is an innovative way to explain the selective benefit of surgical staging and adds pedagogical value to the paper. The writing is clear, cohesive, and in line with the aims of Cancers.

some suggestions

1) The discussion of biomarkers (HPV-ctDNA, OSNA) is highly relevant but somewhat brief and descriptive. Please expand on the potential future role of these biomarkers in predicting para-aortic metastasis or guiding radiotherapy field decisions.

2) Consider citing recent evidence on other promising biomarkers, such as microRNA or methylation markers, which could complement HPV-ctDNA as non-invasive tools for minimal residual disease detection.

3) The manuscript is generally fluent, but a few sentences could be streamlined to improve readability (e.g., the long paragraph starting at line 355 “Surgical staging remains the most accurate method…”).

Author Response

Comment 1
We thank the reviewer for this valuable suggestion. We have expanded the discussion to better contextualize the emerging role of molecular biomarkers—particularly HPV circulating tumor DNA (HPV-ctDNA) and OSNA (One-Step Nucleic Acid Amplification)—in refining nodal risk stratification and treatment individualization. We now discuss how these tools may complement imaging and surgical staging by identifying subclinical para-aortic disease and optimizing radiotherapy field definition in locally advanced cervical cancer.

Change in manuscript: Section 4 – Refining Surgical Staging: From OSNA to Biomarkers, lines 950–1010.

Comment 2
We agree and have expanded the Discussion to include recent evidence on circulating microRNAs and DNA methylation markers as complementary, non-invasive approaches for MRD detection and treatment response monitoring. We now cite studies on exosomal/plasma miRNA panels associated with response to chemoradiotherapy (miR-21, miR-92a, miR-205) and host-cell DNA methylation assays (CADM1, MAL, FAM19A4) with diagnostic and prognostic relevance.

Change in manuscript: Section 4 – Refining Surgical Staging: From OSNA to Biomarkers, lines 1011–1045. Two new references added: Garg et al., BBA Rev Cancer 2024; Dovnik et al., Int J Mol Sci 2023.

Comment 3
We appreciate this stylistic suggestion. The long paragraph in the Conclusions beginning with “Surgical staging remains the most accurate method…” has been revised for clarity and readability. It has been divided into two shorter paragraphs and slightly reworded to improve flow, while preserving the scientific meaning and tone of the manuscript.

Change in manuscript: Section 5 – Conclusions, lines 1150–1170.